# Distinct structural and catalytic roles for Zap70 in formation of the immunological synapse in CTL

Misty R Jenkins[1†], Jane C Stinchcombe[1†], Byron B Au-Yeung[2,3,5], Yukako Asano[1], Alex T Ritter[1,4], Arthur Weiss[2,3,5], Gillian M Griffiths[1]*

[1]Cambridge Institute for Medical Research, University of Cambridge, Cambridge, United Kingdom; [2]Department of Medicine, University of California, San Francisco, San Francisco, United States; [3]Department of Microbiology and Immunology, University of California, San Francisco, San Francisco, United States; [4]Cell Biology and Metabolism Branch, National Institutes of Health, Bethesda, United States; [5]Howard Hughes Medical Institue, University of California, San Francisco, San Francisco, United States

**Abstract** T cell receptor (TCR) activation leads to a dramatic reorganisation of both membranes and receptors as the immunological synapse forms. Using a genetic model to rapidly inhibit Zap70 catalytic activity we examined synapse formation between cytotoxic T lymphocytes and their targets. In the absence of Zap70 catalytic activity Vav-1 activation occurs and synapse formation is arrested at a stage with actin and integrin rich interdigitations forming the interface between the two cells. The membranes at the synapse are unable to flatten to provide extended contact, and Lck does not cluster to form the central supramolecular activation cluster (cSMAC). Centrosome polarisation is initiated but aborts before reaching the synapse and the granules do not polarise. Our findings reveal distinct roles for Zap70 as a structural protein regulating integrin-mediated control of actin vs its catalytic activity that regulates TCR-mediated control of actin and membrane remodelling during formation of the immunological synapse.

*For correspondence: gg305@cam.ac.uk

†These authors contributed equally to this work

Competing interests: The authors declare that no competing interests exist.

## Introduction

T cell receptor (TCR) activation induces a highly structured reorganisation of the receptors at the immunological synapse. The mature synapse forms over several minutes and is segregated into three concentric regions, called the central, peripheral and distal supramolecular activation complexes (SMACs) (reviewed in *Jenkins and Griffiths (2010)*). The regions can be distinguished by the differential localization of molecules. For instance, Lck and PKC-θ cluster with the TCR within the central SMAC (cSMAC), while integrins and associated talin are excluded centrally but accumulate at the peripheral SMAC (pSMAC) (*Monks et al., 1998*) and actin clusters within the distal SMAC (dSMAC). In secretory synapses, formed by cytotoxic T lymphocytes (CTL), secretion occurs into a small secretory cleft next to the cSMAC and within the pSMAC (*Stinchcombe et al., 2001b*). The membranes around the secretory cleft are very tightly opposed around the area where cytotoxic proteins are secreted. Exactly how this organisation of membranes is set up as CTL and target cells interact is not known.

Zap70 is a key kinase in the TCR signalling pathway. Via its tandem SH2 domains, it associates with tyrosine phosphorylated CD3 and zeta chains and is itself phosphorylated by Src family kinases in response to TCR stimulation (*Au-Yeung et al., 2009*; *van Oers et al., 1996*). Zap70 acts as a critical effector of downstream signalling after initial engagement of TCR. Loss of *Zap70* in humans leads to Severe Combined Immunodeficiency (SCID) characterized by the absence of CD8 T cells and the presence of non-functional CD4 T cells (*Arpaia et al., 1994*; *Chan et al., 1994*; *Elder et al., 1994*). Defects in

**eLife digest** White blood cells are responsible for defending the body against infection and disease. Cytotoxic T-lymphocytes, or cytotoxic T cells, are white blood cells that recognise and kill cells that are infected, cancerous or otherwise damaged. Receptors on the surface of these T cells recognise 'foreign' molecules on the surface of diseased or damaged cells: this activates the T cells, which then release cytotoxic proteins that destroy the target cells.

During this process the T cell and the target cell are brought into close contact with each other, and their membranes undergo a dramatic rearrangement to form an 'immunological synapse'. Although the structure of the immunological synapse is understood in detail, the mechanisms controlling the membrane reorganisation are not well understood. Previous studies have shown that an enzyme called Zap70 needs to be present to activate receptors involved in cell adhesion, termed integrins. Now, Jenkins, Stinchcombe et al. have shown a dual role for Zap70 in the formation of the immunological synapse.

Jenkins, Stinchcombe et al. used mice that had been engineered to produce a modified version of Zap70 that worked as normal until its activity was 'switched off' by the addition of a specific drug. When Zap70 was switched off, integrins were still activated but the formation of the immunological synapse was halted with only finger-tip-like contacts between the T cell and the target cell. These contacts were formed by projections from the T cell made of a protein called actin, which forms a kind of scaffolding within cells. Without active Zap70, the T cell receptors could not trigger the dynamic rearrangement of the actin proteins and the membrane remodelling required to form a tight contact between the two cells. This disrupted the delivery of the cytotoxic proteins to their target. These results clearly show that Zap70 has at least two distinct roles that it must carry out for an immunological synapse to form.

thymic development are revealed in mice deficient in *Zap70* where no mature T cells develop due to a block in positive selection (*Negishi et al., 1995*; *Kadlecek et al., 1998*). Due to developmental abnormalities, studies on the role of Zap70 in CTL-mediated killing have been limited.

The derivation of mice expressing an engineered Zap70 mutant, the catalytic activity of which can be blocked by the use of a small molecule inhibitor (*Levin et al., 2008*; *Au-Yeung et al., 2010*) has changed this. This analog-sensitive Zap70 protein [Zap70(AS)] has a methionine to alanine substitution in its catalytic site which allows it to accommodate the bulky ATP-competitive inhibitor, 3-MB-PP1, which impairs Zap70(AS) catalytic function but has little effect on wild-type Zap70. This model, with Zap70(AS) controlled by the addition of a rapidly acting small molecule inhibitor that is genetically selective, has opened the way to studying the role of Zap70 in functional mature T cells. Importantly, this system is able to distinguish the roles played by the catalytic activity of Zap70 as opposed to its structural contributions since the inhibited kinase is present, associates with the TCR, is tyrosine phosphorylated by Lck and has the capacity to recruit other signalling molecules.

Initial studies with this system have shown that, when added to CD4 T cells containing the *Zap70*(AS) allele, 3-MB-PP1 inhibits Zap70(AS) catalytic activity within 30 s, thereby leading to loss of LAT and ERK phosphorylation and ablation of the calcium increase in response to TCR cross-linking. Inhibition of Zap70(AS) does not affect its phosphorylation by the upstream kinase, Lck, nor does it affect the catalytic activity of T cells expressing wild-type Zap70. Intriguingly, studies of CD4 T cells revealed that Zap70 catalytic activity was not required for integrin activation, but rather that Zap70 plays a catalytic-independent role in integrin activation (*Au-Yeung et al., 2010*). The Zap70(AS) model therefore provides an unprecedented opportunity for examining the role of integrin vs TCR mediated events during the formation of the immunological synapse as TCR activation is required for integrin activation (*Burbach et al., 2007*). This is particularly interesting since integrin activation alone has been shown to trigger polarisation of both centrosome and secretory granules to the immunological synapse in Natural Killer (NK) cells; although both activating receptor and integrin activation are required for degranulation (*Bryceson et al., 2005*; *March and Long, 2011*; *Bryceson et al., 2009*) as well as remodelling of synaptic actin required for cytokine secretion (*Brown et al., 2012*). Studies on cytokine secretion from CD4 cells suggest that cdc42 may be involved in actin remodelling at the site of secretion (*Chemin et al., 2012*).

Centrosomal docking at the plasma membrane is a key step for polarised secretion from CTL, delivering secretory granules along microtubules to the point of centrosomal contact at the synapse where cortical actin is reduced at the site of secretion (*Stinchcombe et al., 2006*; *Stinchcombe and Griffiths, 2007*). The importance of centrosomal docking at the plasma membrane is clear in CTL in which Lck expression is conditionally deleted. In Lck-deficient CTL the centrosome only partially migrates towards the synapse and does not reach the plasma membrane; the secretory granules are not delivered to the synapse, and targets are not killed (*Tsun et al., 2011*).

Previous studies have revealed roles for Lck, LAT, SLP76 and Zap70 (*Lowin-Kropf et al., 1998*; *Kuhne et al., 2003*; *Tsun et al., 2011*) in TCR-induced polarisation of the microtubule organising centre (MTOC) towards the immunological synapse. A study using the Jurkat variant, P116, which lacks Zap70, showed only 50% of P116 cells were able to polarise their MTOC towards the synapse after superantigen stimulation (which activates via a LAT-independent pathway [*Bueno et al., 2006*]) while this number increased to 75% if Zap70 was re-expressed in these cells (*Blanchard et al., 2002*). This same study showed that cSMAC recruitment of PKCθ and LAT, was impaired in Zap70-deficient P116 cells, although CD3ζ clustering was not. The link between these events was not clear.

In this study, we analysed the requirements of Zap70 catalytic activity, distinct from its structural role, in the formation of the immunological synapse, polarisation of the centrosome and granules as well as subsequent cytotoxic functions by effector CD8 CTL. We have taken advantage of the Zap70(AS) system in which the structural protein, Zap70, is present but whose catalytic activity can be rapidly and selectively inhibited. This provided us with a unique system in which to ask whether, as in NK cells, integrin activation is sufficient for centrosome and granule polarisation and whether there are distinct roles for the structural functions vs catalytic activity of Zap70 in formation of the immunological synapse.

## Results

### Zap70 catalytic activity is required for CTL mediated killing and cytokine production

C57BL/6 Zap70(AS) CTL were generated by stimulation with irradiated allogeneic Balb/c splenocytes in vitro, each week for 2 weeks, before using the activated CTL for assays. The level of cytotoxicity was determined by lactate dehydrogenase (LDH) release from P815 target cells. When we examined the ability of Zap70(AS) CTL to induce target cell death in the presence of the 3-MB-PP1 inhibitor, we saw a complete abrogation of killing (*Figure 1A*). Given that the inhibition of Zap70 has been shown to impair CD4 T cell activation and cytokine production (*Au-Yeung et al., 2010*), we examined the ability of CTL to produce cytokines after a 5 hr in vitro stimulation with anti-CD3ε. CTL with an inactive Zap70 demonstrated a loss of IFN-γ, TNF-α and IL-2 cytokine production (*Figure 1B*). Therefore, despite being previously activated, CTL still rely on Zap70 signalling for production of cytokines.

### Zap70 is essential for organisation of the immunological synapse structure

We examined the ability of CTL lacking Zap70 catalytic activity, to form immunological synapses. We determined whether they formed cSMAC by looking for the clustering of Lck and PKC-θ at the synapse and whether a pSMAC was formed by assaying their ability to clear the integrin-associated protein, talin, into a concentric ring around the cSMAC. Zap70-inactive CTL were able to bind and form conjugates with target cells almost as well as Zap70-active CTL, with 60% of Zap70(AS) CTL (n = 70) forming conjugates in the presence of 3-MB-PP1 compared with 67% (n = 60) without inhibitor. When activated Zap70(AS) CTL were conjugated to P815 target cells in the presence of 3-MB-PP1, their ability to clear talin into a ring at the pSMAC was impaired (*Figure 2A*). Instead, accumulation of talin labelling was seen across the synapse, when viewed in the z plane (*Figure 2A*, inset). cSMAC formation was also impaired, because the same conjugates displayed a drastic reduction in the accumulation of Lck and PKC-θ at the cSMAC (*Figure 2B,C*). These results indicate that Zap70 activity is important in the redistribution of talin and signalling proteins during the formation of a stable synapse.

Previous data have demonstrated that talin is required for F-actin polarisation to the synapse (*Wernimont et al., 2011*). Therefore, given the loss of talin clearance, immunofluorescence microscopy was used to examine the ability of CTL-P815 conjugates to clear actin and form the distal SMAC (dSMAC) in the absence of ZAP activity (*Figure 3A*). Conjugates were scored based on the phenotypes

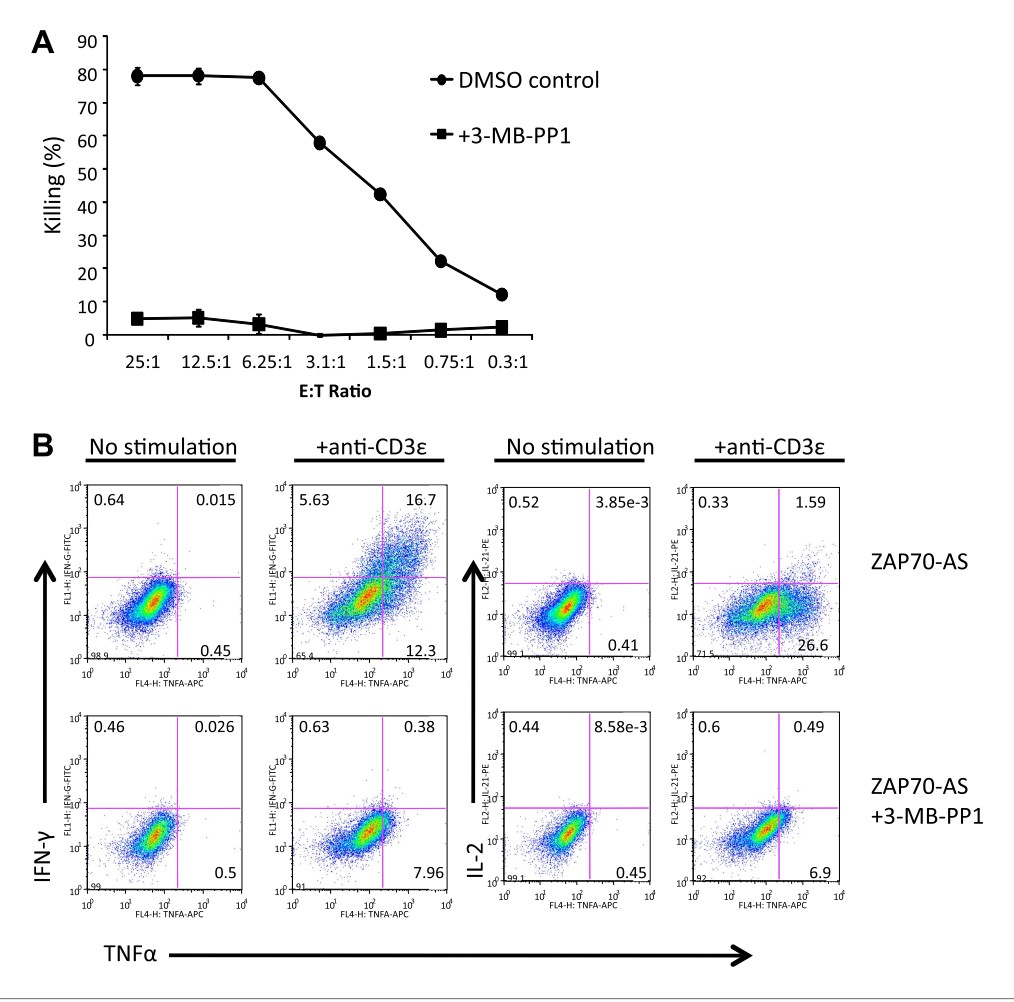

**Figure 1**. T cell killing and cytokine production is dependent on the catalytic activity of Zap70. (**A**) Target cell lysis of P815 targets by Zap70(AS) CTL in the presence (squares) or absence (circles) of 10 µM 3-MB-PP1. Graphs show the mean percentage cytotoxicity of triplicates ± SD for effector to target (E:T) ratios shown; representative of three independent experiments. (**B**) FACS analysis of intracellular staining for IFN-γ (y-axes) TNF-α and IL2 (x-axes) production by Zap70(AS) CTL stimulated with anti-CD3 ±10 µM 3-MB-PP1.

of total actin clearance, partial, or no clearance from the synapse as described in 'Materials and methods' (*Figure 3A*). The majority of conjugates formed with Zap70-active CTL accumulate actin at the contact site, which clears into a ring of actin at the dSMAC of the synapse (76.5%, n = 65) (*Figure 3B*, black) consistent with other studies of CTL (*Tsun et al., 2011*; *Zhao et al., 2012*). In contrast, when CTL lack Zap70 catalytic activity, they lose this ability, and the proportion of conjugates displaying clearance drops to 26% (n = 83) (*Figure 3B*). When actin clearance is examined in conjugates which do not display a cSMAC (75%, n = 83), the effect is even more pronounced, with 91.5% of conjugates failing to clear actin. Interestingly, accumulation of actin at the synapse was still observed in the absence of Zap70 activity (*Figure 3C*). These results reveal that actin accumulates across the synapse in the absence of Zap70 catalytic activity, but fails to clear to form the outer ring or dSMAC. Moreover, cSMAC formation is impaired in the absence of actin clearance.

## Zap70 catalytic activity is required for ERK, but not Vav-1 activation

Previous studies have shown that phosphorylated ERK may play a role in actin reorganisation as ERK co-localises with actin at the immune synapse, is implicated in granule and MTOC polarisation, and is required for CTL degranulation (*Robertson et al., 2005*; *Jenkins et al., 2009*). To determine how the inhibition of Zap70 catalytic activity affects downstream signalling including ERK phosphorylation in

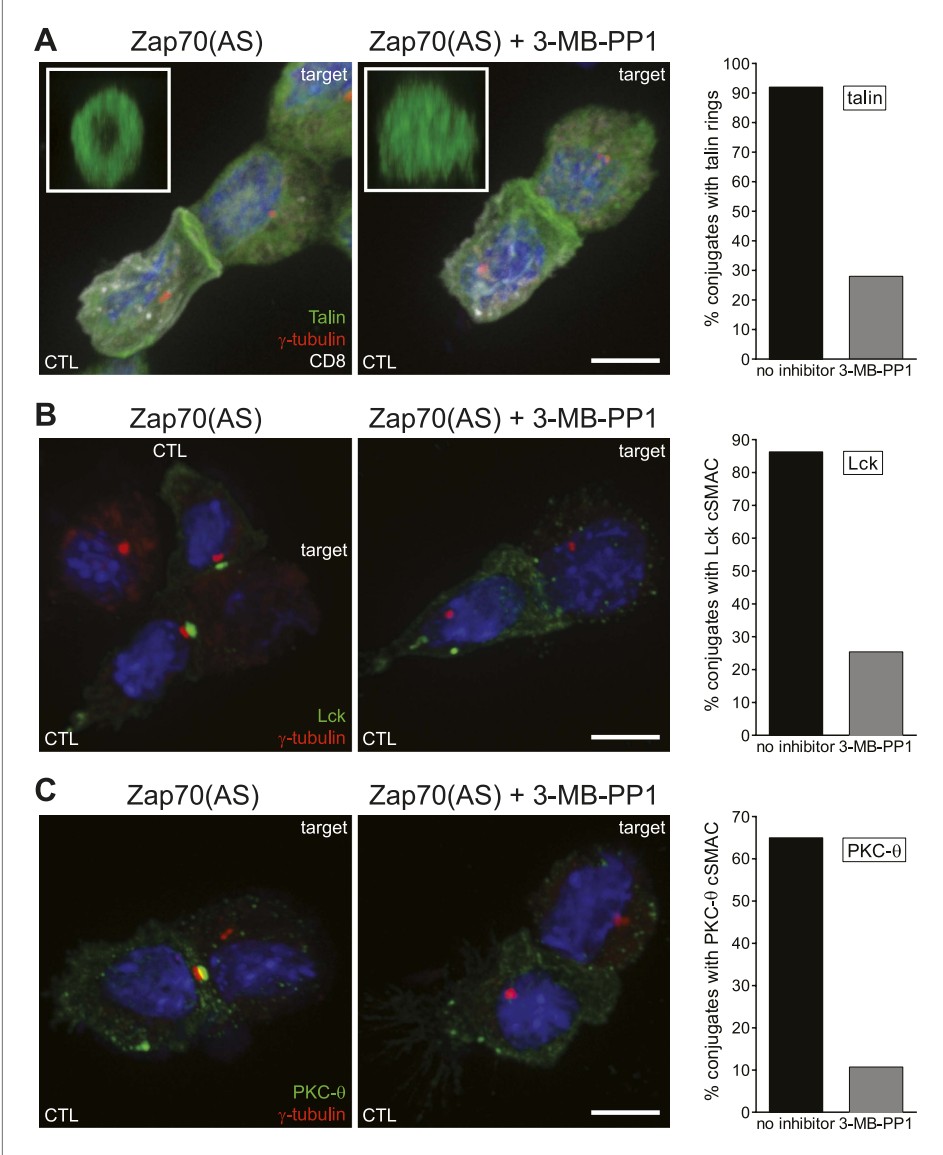

**Figure 2**. Inhibition of Zap70 activity impairs formation of both the cSMAC and pSMAC. (**A**–**C**) Confocal projections of Zap70(AS) CTL conjugated to P815 targets ±10 µM 3-MB-PP1. Cells are labelled with Hoechst (nuclei, blue) and antibodies against γ-tubulin (AlexaFluor 546; red) and either talin (**A**), Lck (**B**) or PKC-θ (**C**) (AlexaFluor-488; green) in the xy plane (scale bar, 5 µm) or as 1 µm reconstructions *en face* across the synapse in the xz plane (insets, scale bar, 3 µm). Graphs show the quantitation of conjugates with the percentages of conjugates displaying talin rings (**A**), Lck cSMACs (**B**) and PKC-θ cSMACs (**C**) at the synapse.

CTL, we performed western blot analysis of *Zap70+/−* and Zap70(AS) CTL lysates (*Figure 4A*). Inhibition of Zap70 catalytic activity does not impair its capacity to be phosphorylated on tyrosine 319, which is inducibly phosphorylated upon CD3 crosslinking, and remains similarly phosphorylated in the presence of a high concentration of 3-MB-PP1 (10 µM). This result is consistent with Zap70 as a substrate of the upstream Src kinase Lck. However, the phosphorylation of LAT, a substrate of Zap70, as well as PLCγ and ERK all depend on Zap70 catalytic activity. These results suggest a signalling pathway from Zap70 via the LAT signalosome and PLCγ is required for activation of ERK in CTL. It has also been reported that a PI3K-dependent pathway is required for activating ERK and CTL degranulation (*Robertson et al., 2005*). To determine whether this pathway may also be dependent on Zap70 activity we probed for phosphorylation of AKT, which is activated downstream of PI3K.

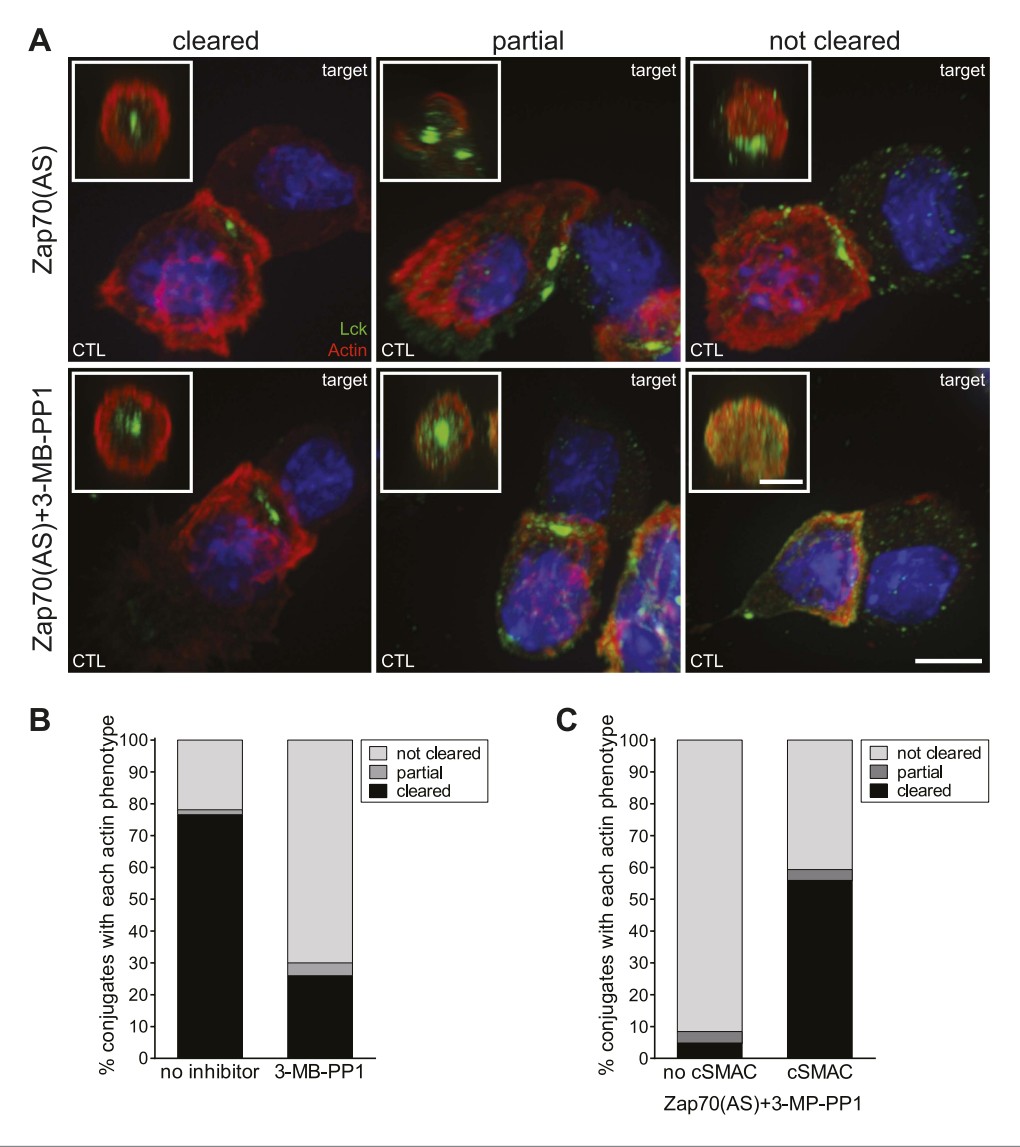

**Figure 3**. Actin clearance from synapses formed by CTL requires Zap70 activity. (**A**) Confocal projections of Zap70(AS) CTL conjugated to P815 targets showing actin organisation at the synapse ±10 µM 3-MB-PP1, in the xy plane (scale bar, 5 µm) or as 1 µm reconstructions *en face* across the synapse in the xz plane (insets, scale bar, 3 µm). Labelling with Hoechst (blue) and antibodies against actin (AlexaFluor-546; red) and Lck (AlexaFluor-488; green). (**B**) Quantitation of actin organisation at the synapse in conjugates in the presence (n = 83) or absence (n = 64) of 10 µM 3-MB-PP1 and (**C**) for conjugates in the presence of 10 µM 3-MB-PP1 (n = 83), in which a cSMAC, identified by Lck clustering, is present (cSMAC) or absent (no cSMAC).

Indeed, Akt phosphorylation was sensitive to Zap70 inhibition, suggesting that ERK is activated downstream of Zap70 through a LAT and PLCγ pathway or alternatively through a pathway that also includes AKT.

Since our previous studies had shown that integrin activation occurs in the absence of Zap70 catalytic activity in naïve CD4 T cells (*Au-Yeung et al., 2010*), we asked whether the integrin-mediated activation of Vav-1 (*Gao et al., 2005*) was seen in CTL in which Zap70 catalytic activity was inhibited. This was of particular interest because Vav-1 has been shown to play an important role in both TCR- and integrin-mediated activation of actin reorganisation in T cells (*del Pozo et al., 2003*, *Garcia-Bernal et al., 2009*; *Tybulewicz, 2005*). We found that TCR activation triggers Vav-1 phosphorylation independently of Zap70 kinase activity, with Vav-1 phosphorylation occurring in both *Zap70+/−* and

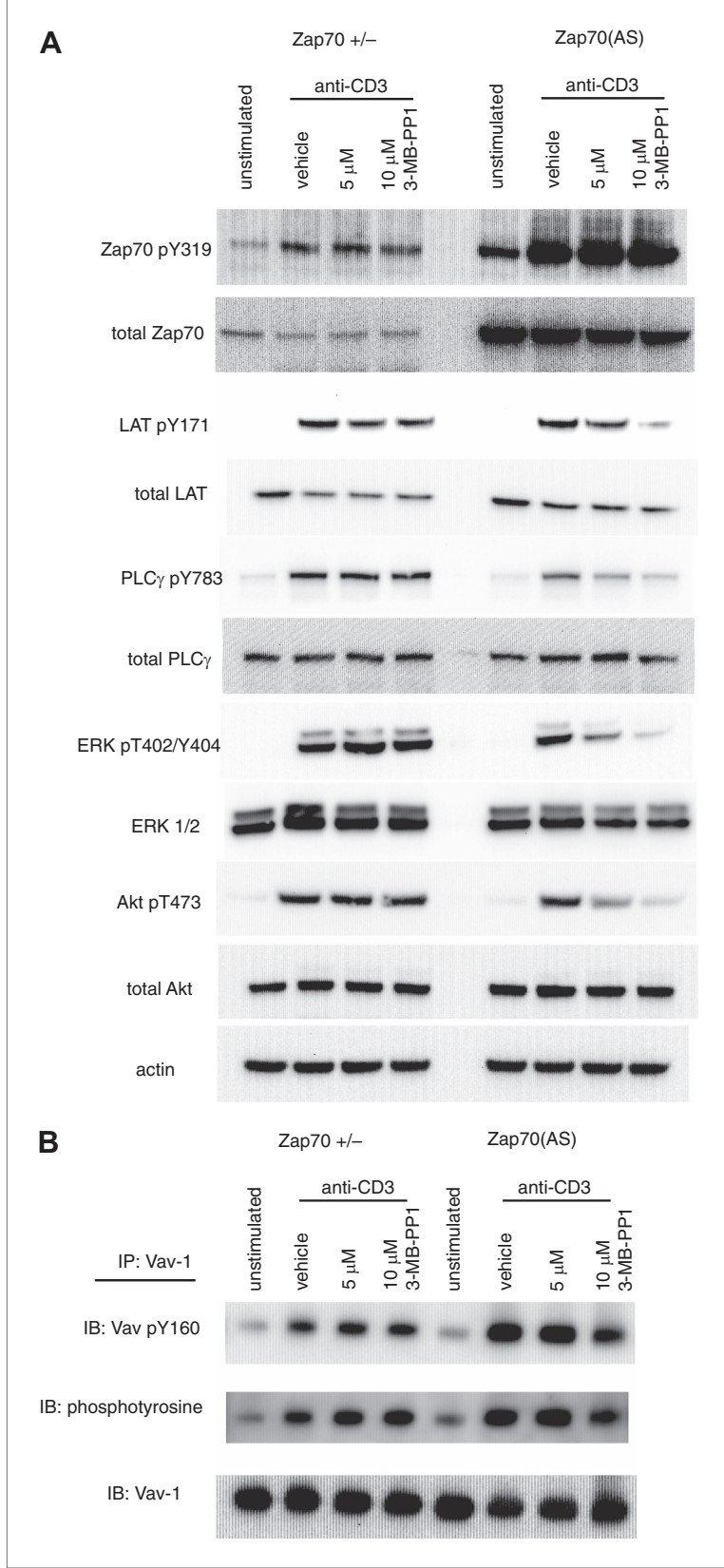

**Figure 4**. TCR signalling downstream of Zap70 is impaired in the absence of Zap70 catalytic activity. (**A**) In vitro generated *Zap70+/−* and Zap70(AS) CTLs were left unstimulated or were stimulated for 2 min by soluble anti-CD3 (10 µg/ml) and cross-linking secondary antibodies, in the presence of vehicle alone (DMSO) or 5 or 10 µM 3-MB-PP1. *Figure 4. Continued on next page*

*Figure 4. Continued*

The phosphorylation status of the indicated TCR signalling molecules was determined by Western blot analysis. Molecular weights: ZAP-70, 70kD; LAT, 38kD; PLCγ 150kD; ERK 42-44kD; Akt, 60kD; Vav-1, 100kD. (**B**) Vav-1 immunoprecipitated from *Zap70+/−* OT-I and Zap70(AS) OT-I CTL treated with vehicle or 3-MB-PP1 as shown. Immunoprecipitates were probed for phosphorylation on tyrosine 160 or total tyrosine phosphorylation of Vav-1.

The following figure supplements are available for figure 4:

**Figure supplement 1**. ICAM adhesion assays.

Zap70(AS) CTL lysates from cells treated with 3-MB-PP1, both for overall Vav-1 phosphorylation as well as at Y160, a site that is selectively phosphorylated upon TCR stimulation or αvβ3 integrin-mediated activation (*Gao et al., 2005*; *Miletic et al., 2006*). Only a slight decrease in Vav1 phosphorylation is evident in Zap70(AS) CTL with 10 µM 3MB-PP1 suggesting Zap70 catalytic function makes only a minimal contribution to Vav-1 phosphorylation (*Figure 4B*). These results suggest that Vav-1 phosphorylation, and presumably its GEF activation, occurs independently of Zap70 catalytic activity, while the pathway leading to ERK and PI3K activation are dependent on Zap70 catalytic activity. This supports the idea that integrin activation occurs in CTL in which Zap70 is catalytically inhibited, as previously observed in Treg (*Au-Yeung et al., 2010*).

We examined integrin activation further by asking whether there were differences in adhesion or in the speed of movement of Zap-inhibited CTL. Adherence to an ICAM-1 coated plate was measured before and after TCR activation using anti-CD3ε for 10 min. TCR activation increased the percentage of CTL binding to ICAM-1 from 20% to 30% for *Zap70+/−* and from 16% to 34% for Zap70(AS) CTL. 3-MB-PP1 treatment resulted in 32% binding for *Zap70+/−* and 25% for Zap70 (AS) CTL (*Figure 4—figure supplement 1*). While there does appear to be some reduction in CTL adhesion upon 3-MB-PP1 treatment, these data also suggest there is residual Zap70 catalytic-independent integrin function. We also examined the level of integrin activation by determining the speed of CTL movement using live cell imaging. We found no difference in the speed of movement on an ICAM-1 surface with CTL moving with an average speed of 9 µm/min when Zap70 was catalytically active or inactive (*Videos 1 and 2*; n>84 each). These results support the idea that Zap70 catalytic activity is not required for integrin activation in CTL. Our finding that conjugate formation is not impaired in Zap70-inactive CTL also supports this model.

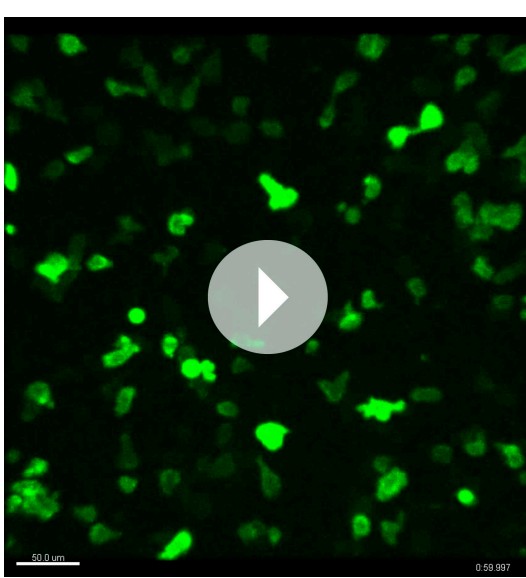

**Video 1**. Live cell imaging of Zap70(AS) OT-I CTL transfected with Lifeact-EGFP (green) moving on a glass coverslip coated with 0.5 µg/ml ICAM−1 + 0.1% DMSO.

Thus, distinct TCR-dependent signalling events are influenced by Zap70 scaffolding and kinase activity. Importantly, catalytically inhibited Zap70 can still participate in signal transduction without active catalysis, by interacting with other signalling proteins via its phosphorylated tyrosines. These results in activated CTL are consistent with previous studies in naïve CD4 T cells, which demonstrated a Zap70 kinase-independent scaffold function, where Zap70 forms a complex with the adapter protein Crk, which activates the GTPase Rap1, which subsequently regulates integrin-mediated adhesion (*Au-Yeung et al., 2010*).

## Zap70 is essential for centrosome and granule polarisation to the synapse

Centrosome polarisation to the synapse is an important step in CTL killing, because centrosome docking at the plasma membrane is responsible for directing the cytolytic granules to the secretory cleft at the cSMAC. Given that NK cells can polarise both centrosome and granules to the synapse in response to integrin activation alone

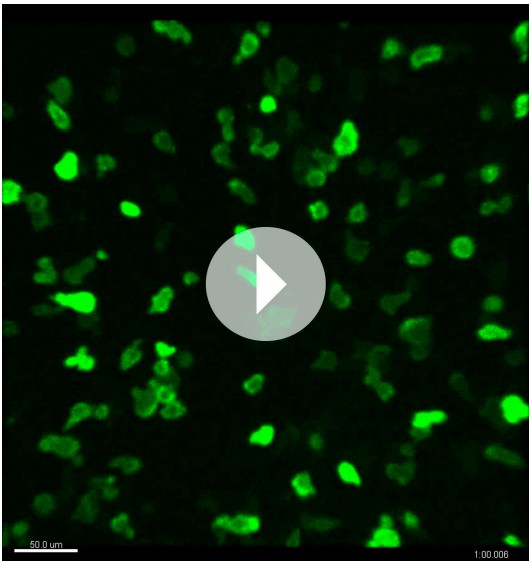

**Video 2**. Live cell imaging of Zap70(AS) OT-I CTL transfected with Lifeact-EGFP (green) moving on a glass coverslip coated with 0.5 µg/ml ICAM−1 +10 µM 3-MB-PP1.

(*Bryceson et al., 2005*), it was of interest to ask whether the centrosome could polarise in CTL in which Zap70 was catalytically inhibited but integrin activation could still occur.

Our previous studies have shown that Lck is essential for docking of the centrosome at the plasma membrane since when Lck expression is inducibly turned off in mature CTL, the centrosome polarises towards the immune synapse, but does not dock at the plasma membrane. In the absence of centrosome docking at the plasma membrane, the granules cannot be delivered to the immunological synapse and CTL killing is abolished (*Tsun et al., 2011*).

Zap70 is a substrate of Lck and its phosphorylation by Lck activates Zap70's kinase activity. Therefore we asked whether centrosome polarisation occurs in the absence of Zap70 catalytic activity by using immunofluorescence microscopy to examine the ability of Zap70(AS) CTL-P815 conjugates to polarise their centrosomes (γ-tubulin labelling) (*Figure 5A*) to the synapse identified by Lck labelling (green) in the presence or absence of inhibitor. Conjugates were classified according to the location of the centrosome relative to the synapse and nucleus (as described in 'Materials and methods' and illustrated in *Figure 5A*), and by the distance between the centrosome and the synapse (*Figure 5B*). The majority of conjugates (65%) with catalytically active Zap70, displayed centrosomes tightly polarised to the cSMAC (*Figure 5B*) (n = 118), in accordance with previously published work (*Jenkins et al., 2009*). In contrast, Zap70 inhibition disrupted centrosome polarisation, with 54% of 3-MB-PP1 treated conjugates showing a centrosome location >5 µm from the synapse (*Figure 5B*) (n = 85).

To determine whether inhibition of Zap70 catalytic activity affected granule polarisation, the Zap70(AS) ±3-MB-PP1 conjugates were labelled with an antibody to LAMP-1 (CD107a) (*Figure 5C*, white), a lysosomal membrane protein used as a marker of the secretory granules in T cells (and lysosomes in the target cells), and scored for the location of the granules, relative to the synapse (as described in 'Materials and methods'). Immunofluorescence microscopy was used to identify granule location relative to the synapse identified by Lck labelling (green) (*Figure 5C*). Although the majority (44%) of control (Zap70(AS)+DMSO) conjugates (n = 117) show granules tightly clustered to the synapse, conjugates treated with 3-MB-PP1 showed a loss of granule polarisation to the synapse (*Figure 5D*) (n = 92). The loss of centrosome and granule polarisation in cells lacking Zap70 catalytic activity is consistent with the loss of cytotoxicity. These results suggest that, in contrast to NK cells, integrin activation is not sufficient for centrosome and granule polarisation to the synapse.

## CTL lacking Zap70 catalytic activity initiate but then abort centrosome polarisation

We also examined centrosome polarisation using live cell microscopy. In order to examine conjugates formed by identical TCR interactions we used CTL derived from lines crossed onto the TCR transgenic OT-I background. Zap70(AS) OT-I CTL were transfected with Lifeact-EGFP (to label polymerised actin) and PACT-mRFP (to label the centrosome) and treated with 3-MB-PP1. Target cell killing by Zap70(AS) OT-I CTL show complete inhibition of killing when treated with 3-MB-PP1 (*Figure 6—figure supplement 1*). Live cell imaging of Zap70-active OT-I CTL show the centrosome polarising right up to the contact site within 6 min of initial interaction with the target (*Figure 6*; *Video 3*; n = 34). However although the centrosome begins to polarise towards the synapse in Zap70(AS) OT-I CTL treated with 3-MB-PP1, the centrosome fails to reach the contact site with the closest point of contact ~3 µm at 6 min after initial contact (*Figure 6*; *Video 4*; n = 50). In each video examined from Zap-deficient CTL, centrosome polarisation began, but aborted before the centrosome docked at the contact site formed by the synapse.

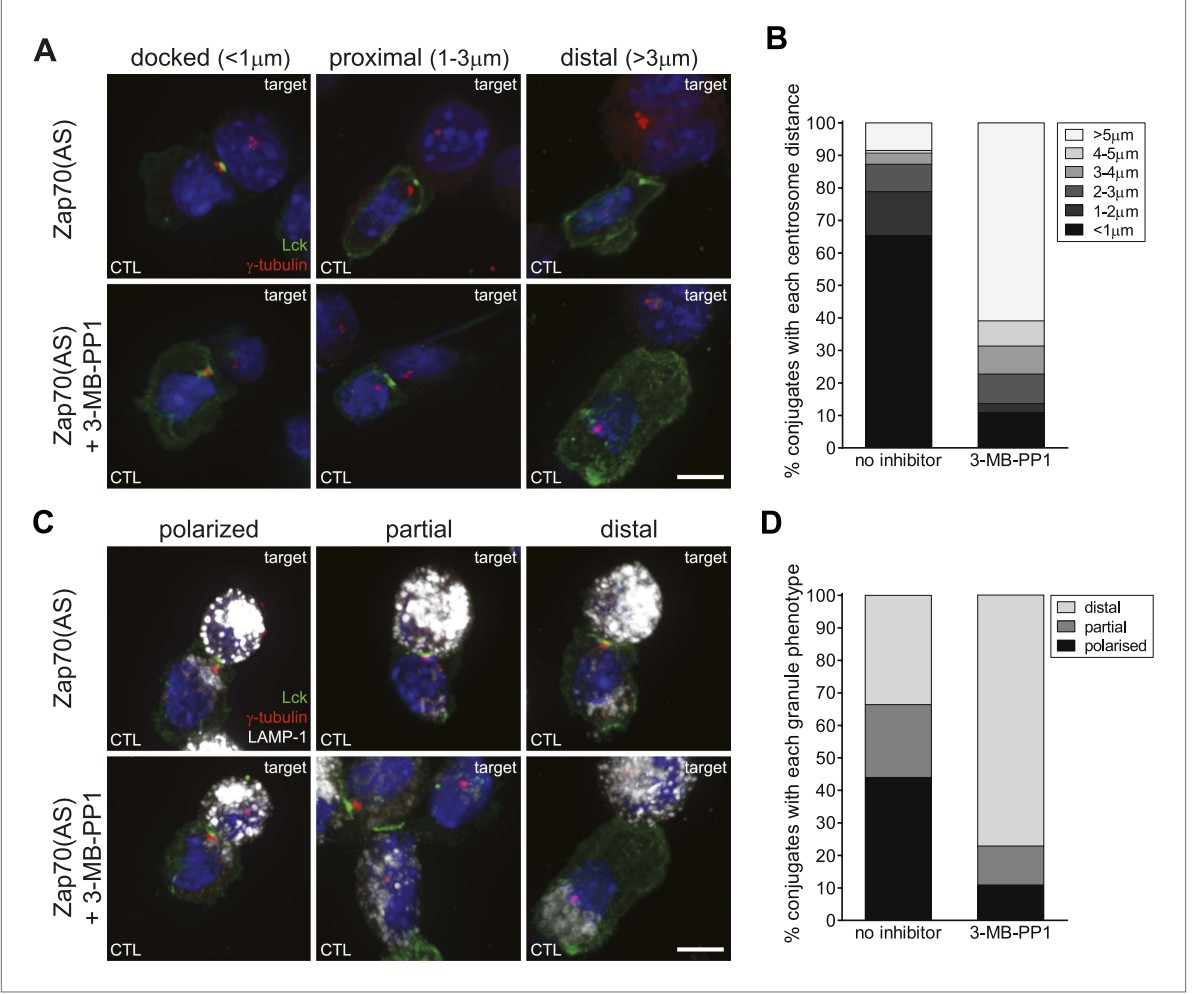

**Figure 5**. Centrosome and granule polarisation to the synapse is impaired in Zap70 inactive CTL. (**A** and **C**) Confocal projections of Zap70(AS) CTL conjugated to P815 targets, labelled with Hoechst (blue) and antibodies against Lck (AlexaFluor-488; green), γ-tubulin (AlexaFluor-546; red) and (**C**) LAMP-1 (AlexaFluor-633; white) (scale bars, 5 μm) illustrating centrosome (**A**) and granule (**C**) polarisation in CTL. Quantitation of conjugates in the presence (n = 325) or absence (n = 163) of 10 μM 3-MB-PP1, showing distance of centrosome from the synapse (**B**) or granule polarisation phenotypes (**D**), illustrated in (**A**) and (**C**), as a percentage of total conjugates formed. (NB LAMP-1 stains both CTL and target lysosomes.)

## Membrane remodelling required for extended contact and secretory cleft formation requires Zap70 catalytic activity

In order to determine the defect in synapse formation and centrosome polarisation and docking at higher resolution we examined the ultrastructure of the contact site formed between CTL and targets when Zap70 catalytic activity was inhibited. Conjugates formed between CTL and targets were fixed and processed for electron microscopy at 25, 40 and 60 min after mixing.

At 25 (*Figure 7A*) and 40 (*Figure 7C*) minutes, the contact sites formed between CTL with active Zap70 and target cells show typical secretory synapses (*Stinchcombe et al., 2001b*) with flat stretches of tight membrane interactions between the cells surrounding a central intercellular gap between the two cell membranes, termed the secretory cleft. The cleft forms an extracellular space containing heterogeneous membranous and granular material. Centrioles, lytic granules and Golgi elements are polarised up to the CTL plasma membrane in the middle of the contact site, opposite the secretory cleft. At these stages of interaction, the target cell appears relatively intact although the endoplasmic reticulum at 40 min is slightly swollen and vacuolated, characteristic of target cell death (*Figure 7C*). By 60 min, there is marked evidence of target cell death with extensive endoplasmic reticular vacuolation and swollen mitochondria, with the CTL appearing to retract from an etiolated, dying target (*Figure 7E*).

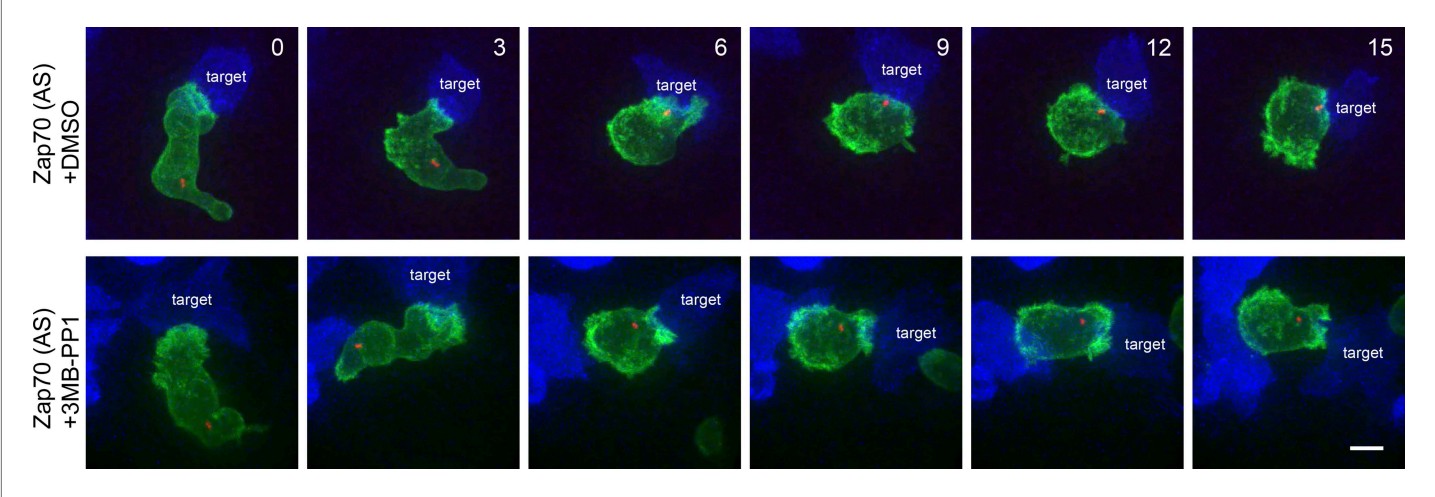

**Figure 6**. CTL lacking Zap70 catalytic activity show abortive centrosome polarisation. Single frames from **Videos 3 and 4** showing 3 min intervals of Zap70(AS) OT-I CTL ±10 μM 3-MB-PP1. CTL were transfected with Lifeact-EGFP (green) and mPACT-RFP (centrosome marker, red), and target cells expressing farnesylated mTagBFP2 (blue). Scale bar, 5 μm; n = 50 for inhibitor and 34 for control treatments.

The following figure supplements are available for figure 6:

**Figure supplement 1**. Lysis of targets by Zap70(AS) OT-I CTL is specifically inhibited by 3-MB-PP1.

In the absence of Zap70 kinase activity the ultrastructure of the synapse is very different (**Figure 7B,D,F**). There is no evidence of the tight, flat membrane–membrane associations between CTL and target. Secretory clefts are absent, and neither centrioles, Golgi cisternae nor lytic granules are polarised towards the target cells. Instead, the contact site is highly interdigitated, formed of long projections from the CTL surface (black arrowheads). Conjugates formed between CTL lacking Zap70 activity and targets after 25, 40 and 60 min of interaction show similar phenotypes. There is no evidence of target cell death at any of these time points in these samples. Notably points of contact between CTL and target are restricted to the tips of projections, providing limited sites for receptor interactions between the cells.

Quantitation of the EM data reveal that with Zap70(AS) CTL + 3-MB-PP1 the secretory cleft is absent in >85% of conjugates analysed at any time point (**Figure 7—figure supplement 1A**); the centrosome is not docked and instead is found at distances greater than 1 μm from the synapse in >70% of conjugates (**Figure 7—figure supplement 1B**) and projections are present between CTL and target (**Figure 7—figure supplement 1C**). In contrast, in the absence of inhibitor Zap70(AS) CTL showed secretory clefts in >50% of conjugates observed at 25 and 40 min, reduced to 27% only at 60 min, when many of the target cells were dead. Tightly polarised centrosomes are seen most frequently at the 25 min time-point when 69% of conjugates have at least one centriole <500 nm from the plasma membrane and 85% of conjugates show no projections between CTL and target. Interestingly up to 60% of CTL lacking Zap70 catalytic activity show the centrosome within 3 μm of the synapse in conjugates formed at 25 min (**Figure 7—figure supplement 1B**), consistent with the initial polarisation up to this point observed in live imaging (**Figure 6**), decreasing to 35% in conjugates formed at 60 min.

The unusual ultrastructural morphology of the synapse formed by CTL lacking Zap70 activity most closely resembles that seen in wild-type CTL at very early stages of synapse formation, when the target cell is intact and before the centrosome has polarised to the synapse (**Figure 7G**). At this stage of synapse formation there are many interdigitations between CTL and target and neither the flattening of the membranes between the two cells, nor the secretory cleft are seen. Here too the contact points between CTL and target are only found at the tips of projections between CTL and target. These observations support the idea that loss of Zap70 catalytic activity arrests the reorganisation of membranes at the synapse at an early stage prior to formation of the secretory cleft.

Importantly these images reveal the ultrastructure that underlies the accumulation of actin seen across the synapses formed by Zap70 catalytically inhibited CTL and at the early stages of synapse

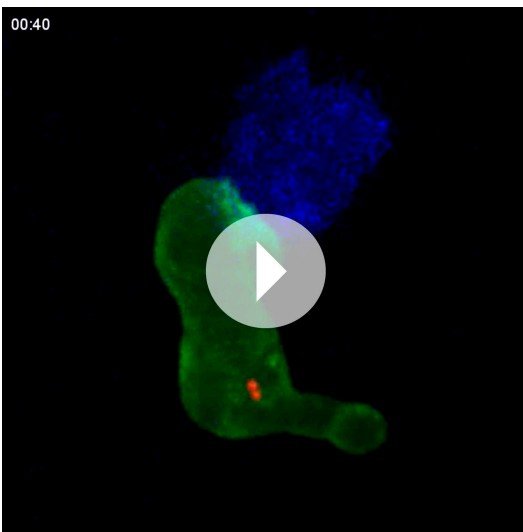

**Video 3**. Live cell imaging of Zap70(AS) OT-I CTL transfected with Lifeact-EGFP (green) and mPACT-RFP (centrosome marker, red), with EL4 target cells expressing farnesylated mTagBFP2 (blue) +0.1% DMSO.

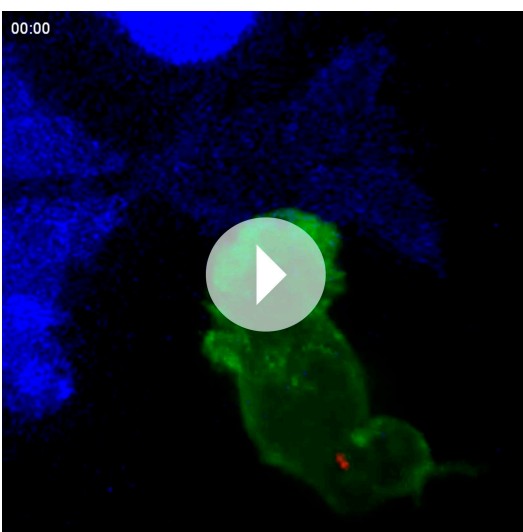

**Video 4**. Live cell imaging of Zap70(AS) OT-I CTL transfected with Lifeact-EGFP (green) and mPACT-RFP (centrosome marker, red), with target cells labelled with EL4 target cells expressing farnesylated mTagBFP2 (blue) +10 µM 3-MB-PP1.

formation in wild-type CTL, revealing the first link between the clearance of actin and the flattening of the membranes to form an extended area of contact during synapse formation.

## Discussion

In this paper we make use of the Zap70(AS) model to study the role of Zap70 catalytic activity in CTL. While earlier studies revealed that target cell killing was inhibited in the absence of Zap70 activity (**Au-Yeung et al., 2010**), the underlying mechanisms for the loss of cytotoxicity were not investigated. We find that in the absence of Zap70 catalytic activity formation of the immunological synapse is arrested at a stage where actin-rich interdigitations dominate the interface between the two cells, with actin accumulated across the synapse and TCR unable to coalesce to form a cSMAC. The membranes between CTL and target are unable to flatten to provide an extended area of contact and the secretory cleft does not form. Signalling downstream of Zap70 is disrupted and neither centrosome nor granules polarise. Our studies provide the first insights into the membrane reorganisations that accompany actin remodelling to form the mature immunological synapse.

The Zap70(AS) mouse offers a unique opportunity to examine the role of the catalytic activity of Zap70 in CTL, which can be inhibited rapidly and specifically by addition of the inhibitor 3-MB-PP1. Previous studies using Zap70(AS) CD4 cells demonstrated that although addition of 3-MB-PP1 inhibits phosphorylation of LAT and ERK, Zap70 itself can still be phosphorylated and take part in signalling. We see the same picture in CD8 CTL. Furthermore we show that phosphorylation of PLCγ, ERK and AKT are all dependent on Zap70 catalytic activity, suggesting that the LAT signalosome and PLCγ are required for ERK activation in CTL, providing a rationale for previous observations implicating both ERK (**Robertson et al., 2005**) and PLCγ (**Le Floc'h et al., 2011**) in CTL-mediated killing. We also find that catalytically inactive Zap70 can itself be phosphorylated in CTL, supporting a kinase-independent scaffolding role for Zap70 in TCR regulation of integrin-mediated adhesion as reported in Treg cells (**Au-Yeung et al., 2010**).

These results support the idea that the initial stages of synapse formation, during which there is an accumulation of actin across the synapse, may well be integrin mediated. Consistent with this is our finding that Vav-1 phosphorylation mediated by the TCR, and possibly involving integrins (**Riteau et al., 2003**; **Garcia-Bernal et al., 2005**, **2009**), appears to be independent of Zap70 catalytic activity, with Y160 phosphorylation occurring in Zap70 catalytically inactive CTL. Although earlier studies have implicated Zap70 in the activation of Vav-1 both in vivo (**Deckert et al., 1996**; **Kadlecek et al., 1998**; **Michel et al., 1998**) and in vitro (**Brunati et al., 1995**; **Han et al., 1998**), our present study shows phosphorylation of Vav-1 in the absence of Zap-70 catalytic activity, likely mediated by Lck.

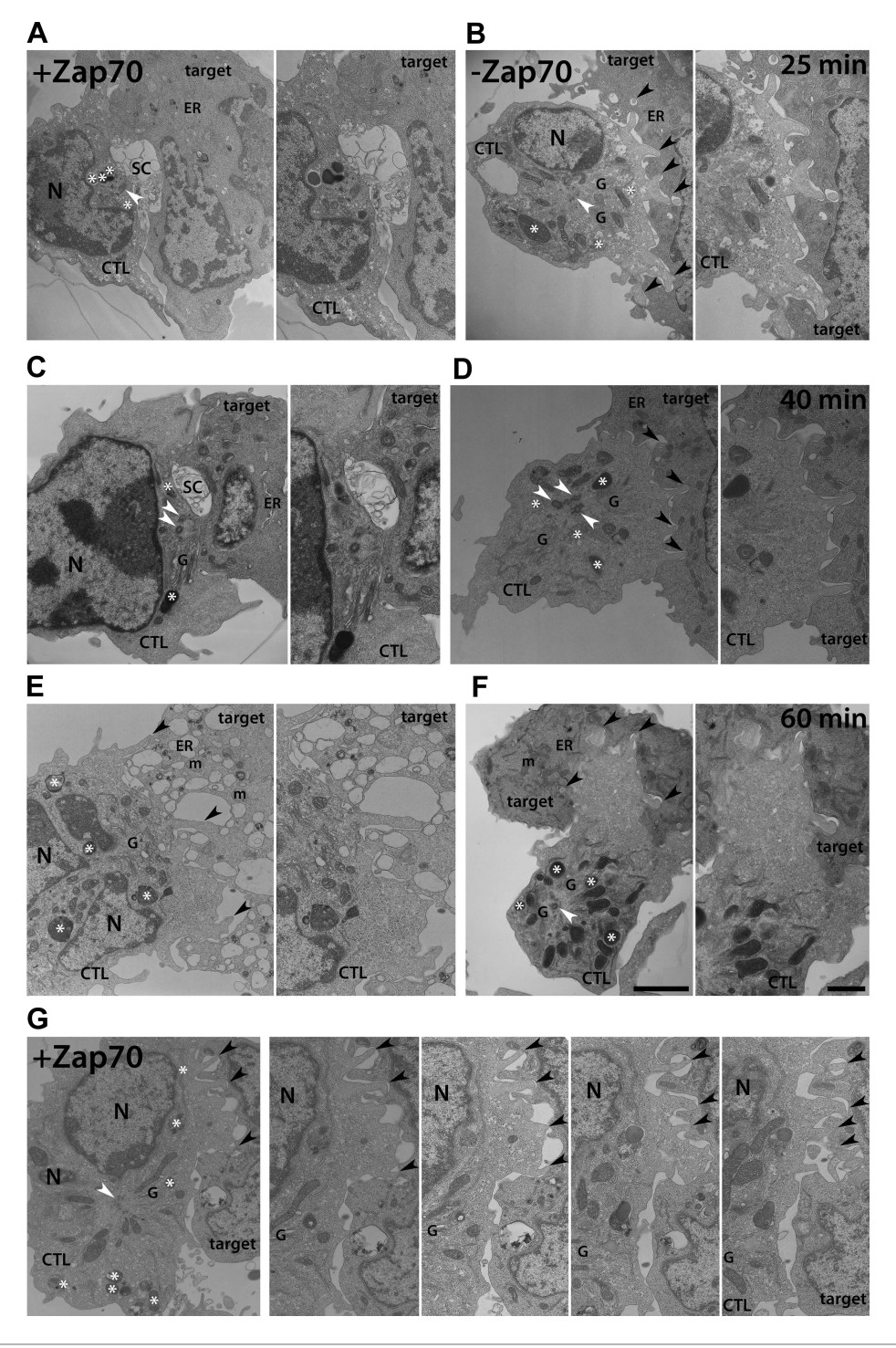

**Figure 7**. The structure of the immunological synapse is severely impaired in CTL upon Zap70 inhibition. Electron micrographs of single (**A**–**F**, left panel **G**) or non-sequential serial (right panels, **G**) thin (50–70 nm, **A**–**E**, **G**) or semi-thin (70–100 nm, **F**) lead-stained sections through the contact site formed between Zap70(AS) CTL (CTL) and P815 target cells (target), conjugated for 25 (**A** and **B**), 40 (**C** and **D**) or 60 (**E**–**G**) min at 37C in the absence (**A**, **C**, **E**, **G**) or presence (**B**, **D**, **F**) of 10 μM 3-MB-PP1. Secretory cleft (SC); interdigitations, between CTL and targets (black arrowheads); centrioles (white arrowheads), lytic granules (asterisks); Golgi elements (G) and nuclei (N) in CTL, and endoplasmic reticular (ER) and mitochondria (m) in target cells, are indicated in lower power images for (**A**–**F**) and *Figure 7. Continued on next page*

*Figure 7. Continued*

all images for (**G**). Scale bars: low power images, 2 μm; high power images bar, 1 μm. Only the ends of the mother centriole appendages are visible in (**B**).
The following figure supplements are available for figure 7:

**Figure supplement 1**. EM quantitation.

---

Our data showing that speed of movement on an ICAM-1 substrate is the same for both Zap70 catalytically active and inactive CTL supports the idea that integrin activation is independent of Zap70 catalytic activity. These results differ from those of an earlier study (*Evans et al., 2011*) using the less specific inhibitor, piceatannol, where the reduced speed of T cell migration may have resulted from inhibition of target kinases other than Zap70 (http://www.kinase-screen.mrc.ac.uk/screening-compounds/345911).

The Zap70(AS) model provides a unique opportunity to ask whether, as in NK, integrin activation alone might stimulate centrosome and granule polarisation to the synapse (*Bryceson et al., 2005*). Our results suggest that CTL differ from NK cells in this respect. Although centrosome polarisation is initiated in CTL lacking Zap70 activity, it is aborted before reaching the synapse, and granule polarisation does not take place. These results suggest an underlying difference in the mechanism of polarisation that may reflect the different roles played by NK and CTL in recognising and destroying virally infected and tumour cells.

In the absence of Zap70 catalytic activity synapse formation is disrupted at both the level of protein reorganisation and ultrastructure. We find that both the integrin-associated talin, as well as actin, accumulate across the immunological synapse, but do not clear to form the concentric pSMAC and dSMAC actin rings characteristic of the synapse. Lck, the kinase that phosphorylates and activates Zap70 and is a marker of the cSMAC, fails to cluster to form a cSMAC. This observation fits with previous observations on P116 Jurkat cells lacking Zap70, which failed to cluster PKCθ and LAT at the synapse (*Blanchard et al., 2002*). Our results suggest that the loss of cSMAC formation arises from a failure of actin to clear from the synapse. Actin has been proposed to act as a 'picket fence' (*Morone et al., 2006*), impeding the movement of proteins laterally within the membrane and an increased density of actin across the synapse might therefore disrupt cSMAC formation by impeding the movement of TCR. In keeping with this model inhibition of actin reorganisation with jasplakinolide on planar lipid bilayers prevents cSMAC formation (*Beemiller et al., 2012*) as well as with earlier observations that actin dynamics are required for effective TCR signalling stemming either from the use of actin inhibitors (*Valitutti et al., 1995*; *Delon et al., 1998*; *Tskvitaria-Fuller et al., 2003*) or depletion of actin regulatory proteins such as dynamin 2 (*Gomez et al., 2005*), or ezrin (*Roumier et al., 2001*), all of which inhibit T cell activation. However our results reveal a much more profound disruption of the whole contact site with actin rich protrusions providing very few points of contact for receptor interactions between the two cells which would impair TCR coalescence to form the cSMAC.

Although it has been known for many years that extensive regions of interdigitations (*Kalina and Berke, 1976*; *Sanderson and Glauert, 1977*, *1979*), flattened areas and secretory clefts (*Bykovskaja et al., 1978*; *Carpen et al., 1982*; *Stinchcombe et al., 2001b*) can form between killer cells and targets, the functional significance of these differences in membrane organisation has not been clear. Our studies reveal temporal and structural links between the accumulated actin and the extensive interdigitations between CTL and target. Comparing the light and EM images reveals that actin-rich interdigitations seen by EM correspond to confocal images in which actin has accumulated across the synapse, while actin 'clearance' in confocal images corresponds to EM images in which the membranes have flattened between the two cells and in which actin-rich protrusions are only apparent at the edges of the synapse reaching out around the target (*Figure 7A,C*). Consistent with our observations, Ueda et al noted 'invasive pseudopodia' forming interdigitations between CD4 T cells and B cells at early stages of interaction (*Ueda et al., 2011*).

Our data reveal that, in the absence of Zap70 catalytic activity, the reorganisation of the membranes between CTL and target does not take place. The secretory cleft is also notably absent from synapses formed by Zap70 catalytically-inactive CTL. Although one possible explanation for loss of the secretory cleft, into which granules secrete their content, could be that this cleft results as a loss of secretion

from these cells, this seems highly unlikely as the cleft forms properly in synapses made by secretion-deficient CTL lacking Rab27a or Munc13-4 (*Stinchcombe et al., 2001a*, *2004*). The very striking loss of the cleft structure in synapses formed by Zap70 inactive CTL demonstrates that cleft formation requires Zap70 activity.

In the absence of Zap70 activity, downstream TCR signalling is severely impaired, Lck does not cluster to form a cSMAC and although the centrosome begins to polarise it does not reach the synapse. With the loss of centrosome migration, the lytic granules remain dispersed within Zap70 inactive CTL, do not polarise to the synapse and the target cell is not killed. Previous studies with CTL lacking Lck showed that the centrosome was able to polarise around the nucleus towards the synapse, but was unable to dock at the plasma membrane (*Tsun et al., 2011*). Since live cell imaging was not used to examine centrosome polarisation in Lck-deficient CTL, a direct comparison is not possible. Interestingly actin and talin clearance from the synapse were also impaired in Lck-deficient CTL.

Overall our data support a two-stage model in which the initial accumulation of actin at the synapse is mediated by TCR-facilitated integrin activation (inside out activation of integrins), in which Zap70 plays a scaffolding role (*Au-Yeung et al., 2010*), but the subsequent reorganisation of actin requires Zap70 catalytic activity. Integrin-mediated actin dynamics proceed normally in Zap-inactive CTL. Not only is Vav-1 phosphorylated, but actin accumulates at the synapse and our live cell studies show that actin reorganisation required for integrin-mediated motility is unimpaired in the absence of Zap70 activity. The loss of Zap70 catalytic activity does not affect the integrin driven accumulation of actin at the synapse, but results in loss of actin reorganisation once the synapse has formed.

Our studies reveal a link between the accumulation of actin and the extensive actin-rich interdigitations formed between CTL and target. It is interesting to note that the only points of contact between CTL and target are at the tips of the interdigitations, and TCR proteins localised to these tips would appear as microclusters. When Zap70 is catalytically active actin clears centrally, the membranes flatten into an extended area of contact and TCR microclusters coalesce to form the cSMAC. Our data are consistent with studies showing that actin clearance is required for the coalescence of microclusters to form the cSMAC (*Campi et al., 2005*; *Babich et al., 2012*; *Beemiller et al., 2012*; *Yi et al., 2012*) and we propose that this marks a transition from a highly interdigitated to a flattened interface between the two cells.

In this study we find that inhibition of Zap70 catalytic activity arrests the progress of synapse formation at an early stage. We find that, unlike NK cells (*Bryceson et al., 2005*), CTL are unable to polarise their centrosome and granules in response to integrin activation alone. Our findings also reveal a surprising new role for Zap70 in controlling the reorganisation of membranes to form the interface at the immunological synapse, including the cSMAC and secretory cleft. Without these rearrangements, the synapse is not functional. Our findings point to distinct roles played by Zap70 as a structural protein regulating integrin-mediated control of actin vs the role of it's catalytic subunit controlling TCR mediated control of actin and membrane remodelling during formation of the immunological synapse.

## Materials and methods

### Generation of mouse CTL and cell culture

Single-cell suspensions of naive splenocytes were generated using a 70 µM nylon strainer (Becton Dickinson). Equal numbers of Zap70(AS) or *Zap70+/−* responders were stimulated with BALB/c-derived stimulator splenocytes (irradiated at 3000 Rad) in RPMI 1640, 10% FCS, L-glutamine, sodium pyruvate, 50 U/ml penicillin/streptomycin (Gibco), 50 µM β-2-mercaptoethanol with 100 U/ml human recombinant IL-2 (Roche, UK) (c-RPMI), and cultured at 37°C and 5% $CO_2$. After 4 days, CTL were purified by separation over Ficoll Histopaque 1083-1 (Sigma–Aldrich, UK), washed three times and resuspended in c-RPMI, at $1 \times 10^6$ cells/ml. CTL were stimulated every 7 days, up to four times. P815 and EL4 mouse target cells were maintained in RPMI, 10% FCS and L-glutamine. Zap70(AS) OT-I or *Zap70+/−* OT-I were activated for 4 days with SIINFEKL as previously described (*Jenkins et al., 2009*).

### Reagents, antibodies and western blotting

3-methylbenzyl-pyrazolopyrimidine (3-MB-PP1) was synthesised as described (*Levin et al., 2008*) and a stock solution at 10 mM (1000x) was dissolved in DMSO. Antibodies used for immunofluorescence studies were: mouse anti-actin (AC-40), rabbit anti-actin and rabbit anti-γ-tubulin (Sigma–Aldrich, United Kingdom); Mouse anti-mouse Lck (3A5) (Millipore, United Kingdom); rat anti-mouse CD8 (YTS192) (gift from H Waldmann, Oxford University); rat anti-mouse CD107a (LAMP-1, 1BD4) (Developmental

Studies Hybridoma Bank, University of Iowa, Iowa City, IA) and all secondary Alexa Fluor antibodies (405, 488, 546 and 633) were obtained from Invitrogen. Western blot analysis of CTL lysates was performed as previously described (*Au-Yeung et al., 2010*) with antibodies obtained: rabbit antibodies against phospho-ERK T202/Y204, Zap70 Y319, LAT Y171, AKT S473, total LAT and total AKT (Cell Signaling, Danvers, MA); rabbit anti-PLCγ Y783 (Biosource, San Diego, CA); anti-phosphotyrosine (4G10), and anti-PLCγ (total) (Millipore); anti-ERK 1, ERK 2, and Vav-1 (Santa Cruz, Santa Cruz, CA), Vav Y160 (R&D Systems, Cambridge, MA) and anti-β actin (Sigma, Ronkonkoma, NY). Immunoprecipitations were performed by stimulation of CTL with soluble anti-CD3ε and goat anti-Armenian hamster (Jackson Immunoresearch, West Grove, PA) crosslinking antibodies for 2 min, followed by centrifugation and resuspension of cells in 1% NP-40 alternative lysis buffer with protease inhibitors. Vav-1 was immunoprecipitated with anti-Vav-1 (Santa Cruz) coated protein G sepharose beads (GE Healthcare, Pittsburgh, PA) and eluted from the beads with SDS sample buffer containing 1% dithiothreitol.

## Immunofluorescence and confocal microscopy

Zap70(AS) CTL (5–8 days after the 3rd or 4th stimulation) and P815 target cells were washed in RPMI, resuspended at $4 \times 10^6$ cells/ml and mixed 1:1, with 10 µM 3-MB-PP1 or DMSO-only control and incubated in suspension for 5 min, before aliquoting onto glass multi-well slides (Hendley), and incubated for a further 15 min at 37°C. Samples were then placed on ice and fixed for 5 min with −20°C methanol, washed six times in PBS and blocked in blocking buffer (PBS +1% BSA [Sigma]). Primary antibodies were resuspended in PBS +0.2% BSA and incubated for 1 hr at room temperature, or overnight at 4°C. Samples were washed extensively in PBS +0.2% BSA before adding secondary antibodies for 40 min at room temperature. Nuclei were stained with Hoechst (1:10,000) in PBS for 5 min before mounting with 1.5 coverglass and Mowiol. Samples were examined using the Andor Revolution Spinning Disk microscope (with an Olympus microscope, 100x objective) and lasers exciting at 405, 488, 543 and 633 nm.

## Live cell imaging

CTL derived from Zap70(AS) OT-I and *Zap70+/−* OT-I were nucleofected with PACT-mRFP (*Gillingham and Munro, 2000*) and Lifeact-EGFP using 5 million cells/nucleofection and the mouse T cell nucleofector kit (Lonza, Germany) with 1 ml mouse T cell nucleofector medium with Component B (Lonza) added post-nuclefection. CTL were topped up with c-RPMI, split between three wells of a 12-well plate and used for imaging 24 hr post nucleofection. EL4 target cells expressing farnesylated mTagBFP2 were pulsed with 1 µM SIINFEKL peptide for 40 min and washed in DMEM to remove peptide and serum, then resuspended in DMEM at $6.5 \times 10^5$ cells/ml. 250 µl of these target cells were aliquoted onto the glass coverslip of 35 mm glass bottom culture dishes (MatTek, Ashland, MA) pre-coated overnight with 0.5 µg/ml ICAM-1/FC (R&D Systems, Minneapolis, MN) at 4°C. Targets were allowed to settle and adhere for 5 min, after which 1.5 ml of RPMI lacking phenol red (Gibco, United Kingdom) supplemented with 10% FCS, 25 mM HEPES, and 50 U/ml penicillin and streptomycin (Imaging medium) were added to each chamber with 10 µM 3-MB-PP1 or 0.1% DMSO. Approximately 2 million CTL were centrifuged at 1200 rpm for 4 min and resuspended in 280 µl of imaging medium with 10 µM 3-MB-PP1 or 0.1% DMSO. CTL were added drop-wise over target cells before imaging was started.

CTL-target interactions were imaged at 37°C using an Andor (United Kingdom) Revolution Spinning Disk microscope with 20x or 100x objective, 1.2x camera adapter and environmental chamber (Oko-lab, Japan) for temperature and $CO_2$ regulation. Serial confocal 0.8 µm Z-stacks were taken at 20 s intervals with excitation of 405 nm, 488 nm, and 561 nm at each Z plane. Videos were processed and analysed using Andor iQ2 software (Andor Technologies, United Kingdom), Imaris x64 (Bitplane, Switzerland) and ImageJ (NIH, USA).

## Quantitation of light microscopy

Conjugate formation was assessed by counting the percentage of CD8 labelled cells conjugated to targets, with all cell nuclei labelled with Hoechst. cSMAC formation was determined by Lck or PKC-θ clustering at the synapse. Clusters were scored when >80% of Lck or PKC-θ in the cell was clustered in the centre of the synapse. Actin rings were classified as 'cleared' when a contiguous ring of actin formed the dSMAC; any interruption in this ring was scored 'partial' and when a solid wall of actin was observed across the synapse this was scored as 'not cleared' as illustrated in *Figure 3*. Centrosome position was determined by measuring the distance between the γ-tubulin labelled centrosome and the Lck or PKC-θ labelled cSMAC or contact site using Imaris software (Bitplane). Centrosomes

were categorised as 'docked' when the centrosome was <1 µm from the synapse; proximal when on the synapse side of the nucleus within 1–3 µm and distal when >3 µm from the synapse. Granules were scored as polarised when >75% of granules were clustered within 5 µm of the synapse; partial when >50% of granules were within 5 µm of the synapse and distal when >75% of granules were >6 µm from the synapse.

## Electron microscopy

CTL were taken 5–6 days after the 2nd or 3rd stimulation and incubated overnight in the presence of 1 mg/ml horseradish peroxidase, (HRP) (Serva, Germany) added directly to the growth medium, to load the secretory lysosomes via the endocytic pathway. Cells were washed extensively in RPMI to remove serum and residual HRP from the medium, resuspended in RPMI to a final concentration of $1–2 \times 10^6$ cells/ml and mixed 1:1 with P815 targets (pre-washed and resuspended to $1–2 \times 10^6$ cells/ml as above) ±10 µM 3-MB-PP1. Cells were left in suspension at room temperature for 5 min, after which they were mixed gently (by pipetting up and down), plated in 4-well tissue culture dishes (Nunc) at 0.5 ml/well and incubated at 37°C for a further 20, 35, or 55 min before fixation with 2% paraformaldehyde and either 1.5% or 3% gluteraldehyde (*Stinchcombe et al., 2001b*). Samples were further processed for DAB cytochemistry, osmium fixation and urynal acetate staining and EPON embedding as previously described (*Stinchcombe et al., 2001a*, *2001b*, *2011*; *Jenkins et al., 2009*). Thin (50–70 nm) and semi-thin (100–150 nm) sections were stained with lead citrate and viewed using a Phillips C100 TEM (FEI). Images were captured using Kodak photographic negative film (Kodak, United Kingdom) and digital electron micrographs produced using a Flextight X5 scanner (Hasselblad, United Kingdom).

## Quantitation of EM micrographs

Thin sections prepared from samples of Zap70(AS) CTL conjugated to targets for 25, 40 or 60 min ±10 µM 3-MB-PP1, were stained with lead citrate and imaged using a Phillips C100 TEM microscope. Images of CTL-target conjugates in which at least one centriole was present were analysed, with 22–60 conjugates for each time point ±10 µM 3-MB-PP1. The secretory cleft was defined as a clear gap within the centre of the contact site between the CTL and target, bounded by an area of tight, flat, membrane–membrane contact on each side. Centrosome distance was measured as the shortest distance from the point of the centriole closest to the plasma membrane and the synapse membrane itself. Measurements were classified as < 500 nm (i.e., one centriole-barrel length, tightly associated), 500–1000 nm (polarised), or >1000 nm (unpolarised) from the plasma membrane. The contact site was defined as the distance between the furthest points of contact between CTL and target at the synapse. The number of membrane extensions projecting from the surface of each CTL with a visible centriole within the contact site was recorded for each CTL condition. Only extensions >1000 nm were included in this analysis.

## Cytotoxicity and degranulation assays

Cytotoxicity was examined using the CytoTox 96 Non-Radioactive Cytotoxicity Assay (Promega, United Kingdom). P815 target cells were resuspended in phenol-free RPMI, 2% FCS at $10^5$ cells/ml in a round bottom 96-well plate. CTL ±10 µM 3-MB-PP1, were added at effector:target (E:T) ratios shown, and plates were incubated at 37°C for 4 hr. The absorbance of the supernatants at 490 nm determined the release of lactate dehydrogenase and % target cell lysis.

## Intracellular cytokine staining

Zap70(AS) CTL were cultured with or without hamster anti-CD3ε (145-2C11, Becton Dickinson Bioscience, United Kingdom) for 5 hr at 37°C in 96-well round bottom plates at approx $0.5–2 \times 10^6$ cells/well in c-RPMI medium containing 5 µg/ml GolgiPlug (Becton Dickinson) ±10 µM 3-MB-PP1. The cells were then washed with PBS (containing 0.1% BSA and 0.02% sodium azide), stained with anti-mouse CD8α-PerCPCy5.5 (Pharmingen) for 30 min on ice, permeabilised by paraformaldehyde fixation using the BD Cytofix/Cytoperm Kit (Becton Dickinson) and stained for intracellular cytokine production using anti-mouse IFNγ-FITC (clone XMG1.2), anti-mouse TNFα (clone MP6-XT22), and anti-mouse IL-2 (clone JES6-5H4) (Pharmingen, United Kingdom). Lymphocytes were washed and analysed on a FACScalibur and analysed using CellQuestPro software (Becton Dickinson). In each assay, any cytokine positive cells isolated from wells with no were subtracted from the % cytokine positive cells incubated with peptide to yield the final value.

## ICAM adhesion assay

CTL ($5 \times 10^5$ per well) were stimulated in triplicate wells for 10 min with 5 µg/ml soluble anti-CD3e (clone 145-2C11) and 50 µg /ml crosslinking goat anti-Armenian hamster (Jackson Immunoresearch) Abs in 96 well plates coated with 10 µg/ml recombinant ICAM-1-Fc (R&D Systems). Cells were washed three times with pre-warmed media, removed with cell dissociation buffer (Gibco) and plate-bound cells were counted by FACS. The percentage of cells bound was calculated as ([number of live CD4 cells bound to the plate) ÷ (input number of cells per well]) × 100.

## Measurement of cell speed

Zap70(AS) OT-I and *Zap70+/−* OT-I CTL were nucleofected with LifeAct-eGFP and imaged as described in 'Live cell imaging' above using the 1.2x camera adapter x20 objective lens. Serial confocal 1.6 µm 10 planes of Z-stacks were taken at 20 s intervals for 15 min (= 45 time points) with excitation of 488 nm. Videos were analysed using the 'Spots module' of Imaris software (7.6.0 Bitplane) to detect each cell as a spot. Only cells with track durations >140 s and track displacement lengths >13 µm were analysed in order to exclude immobile cells. The mean speed for each set of conditions was calculated from 40–50 cells per condition.

## Acknowledgements

We thank Chao Zhang and Kevan Shokat for synthesizing and providing 3-MB-PP1, and Winnie Lui-Roberts for assistance with preparing the figures.

## Additional information

### Funding

| Funder | Grant reference number | Author |
| --- | --- | --- |
| Wellcome Trust | 075880, 100140 | Gillian M Griffiths |
| Howard Hughes Medical Institute | | Arthur Weiss |
| Australian National Health and Medical Research Council (NHMRC) Biomedical Fellowship | 567082 | Misty R Jenkins |
| Arthritis Foundation | 5476 | Byron B Au-Yeung |
| NIAMS, National Institutes of Health | RC2AR058947 | Arthur Weiss |

The funders had no role in study design, data collection and interpretation, or the decision to submit the work for publication.

### Author contributions

MRJ, JCS, BBA-Y, YA, ATR, Acquisition of data, Analysis and interpretation of data, Drafting or revising the article; AW, GMG, Conception and design, Analysis and interpretation of data, Drafting or revising the article

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
