## [Decision Letter]

Thank you for sending your work entitled “Distinct structural and catalytic roles for Zap70 in formation of the immunological synapse in CTL” for consideration at *eLife*. Your article has been favorably evaluated by a Senior editor, a Reviewing editor, and 2 reviewers.

The Reviewing editor and the two reviewers discussed their comments before we reached this decision, and the Reviewing editor has assembled the following comments to help you prepare a revised submission.

The manuscript by Jenkins et al. describes the role of the kinase activity of Zap70 in synapse formation and function in cytotoxic CD8+ T lymphocytes. The authors use the Zap70(AS) mouse whose T cells express a modified form of Zap70 that can be catalytically inactivated upon addition of the chemical 3-MB-PP1. This genetically modified model, described by two of the manuscript co-authors in 2010, allows distinction between T cell functions that require Zap70 as a scaffold protein and others that need its enzymatic activity. Previous experiments using this model showed that Zap70 plays a kinase-independent role in integrin activation upon CD4+ T cell activation, providing a unique system to distinguish between integrin-dependent/TCR-independent and /TCR-dependent events. In the present study, the authors use CTLs generated from Zap70(AS) mice to characterize the role of integrins versus the TCR in synapse formation and function. Overall, the manuscript is clearly written and the results are very clear-cut and interesting. This manuscript should make an important contribution to the field if the authors address the few concerns raised below:

1) The authors have quantified their experiments by counting the number of cells that display or not the analyzed phenotype. However, the quality of their study would be greatly improved if they could further quantify their images using non-biased image analysis software. An important amount of information is lost when the data are analyzed as they do. For example: why do they count the number of polarized cells rather than measure the distance between their centrosome and the synapse? This measure could be indicative of the mechanisms involved in Zap70 kinase-dependent regulation of centrosome polarization.

Time should also be added on the movies in order to have an idea of the speed of displacement of both control and Zap70 inactivated cells.

2) In their Abstract, the authors write, “Unlike Natural Killer cells, which polarise both centrosome and granules in response to integrin activation alone, CTL do not do so”. Yet, this assumption, which is widely adopted in the Discussion, does not rely on experimental data showing that integrin activation is normal in CTL from Zap70(AS) mice treated with 3-MB-PP1. Besides, the movies that have been performed on ICAM-1 coated slides (unlike the fixed microscopy that was performed on uncoated slides) suggest that treated CTL do not migrate the same way on ICAM-1 alone. This would corroborate the results published by N. Hogg's group (The integrin LFA-1 signals through Zap70 to regulate expression of high-affinity LFA-1 on T lymphocytes. Evans R, Lellouch AC, Svensson L, McDowall A, Hogg N. Blood. 2011 Mar 24;117(12):3331-42. Not cited by the authors), which show that ZAP70 kinase activity controls the speed of migration of T cells on ICAM-1 as well as their firm adhesion. Thus another interpretation of the data is the following: Zap70 kinase activity controls LFA1 activation in T cells and firm adhesion of CTL to targets. In the absence of LFA1 activation remodeling of actin at the synapse does not take place as shown in NK cells by Brown AC and co-workers (Super-resolution imaging of remodeled synaptic actin reveals different synergies between NK cell receptors and integrins. Brown AC, Dobbie IM, Alakoskela JM, Davis I, Davis DM. Blood. 2012 Nov 1;120(18):3729-40. Not cited by the authors).

The authors should prove that LFA-1 activation in their model is normal.

3) The authors stress the fact that Zap70 is required as a scaffold protein but not as a kinase for Vav1 activation and formation of actin protrusion. However, the data they provide are not sufficient to state that phosphorylation is normal since phosphorylation of specific sites has not been investigated. Moreover, Vav1 might be phosphorylated but non-correctly localized at the immune synapse. They should therefore either provide the needed data or moderate their conclusion.

4) The main difference in the time-lapse movie shown in Figure 6 is that whereas the *Zap70*^*+/-*^ T cell is static, the one that has lost Zap70 activity is moving around. Does Zap70 kinase activity regulate the formation of synapses versus kinases? Is there any significant change in motility between both cell types?

5) In their Abstract, the authors write, “The membranes at the synapse are unable to flatten to provide extended contact, and TCR microclusters are unable to coalesce.” Yet, none of their data demonstrate this statement: Neither measurement of TCR/CD3 enrichment at the synapse nor analysis of TCR microcluster movements is performed in this study. This statement should thus be removed from the Abstract and if stated in the paper presented as a hypothesis.

---

## [Author Response]

*1) The study would be greatly improved with further quantitation of the images*.

This is a good point and we have provided the quantitation requested, measuring centrosome distances from synapse and times on the single frames from the movies.

2) *The authors should prove that LFA-1 activation in their model is normal*.

We have examined the activation of integrins in our model in three ways: (i) by measuring the adhesion of cells; (ii) by measuring the speed of movement of the cells on an ICAM-1 surface and (iii) by examining both total and site specific (Y160) phosphorylation of Vav1. Taken together these three approaches indicate that integrin activation is normal in Zap70-catalytically inactive CTL compared to controls.

*…the movies that have been performed on ICAM-1 coated slides (unlike the fixed microscopy that was performed on uncoated slides)*…

In fixed imaging ICAM-1 interactions are provided by the target cells themselves, with conjugates formed in suspension before plating.

*Discussion of Evans R, Lellouch AC, Svensson L, McDowall A, Hogg N. Blood. 2011 Mar 24;117(12):3331-42 and Brown AC*, *Dobbie IM, Alakoskela JM, Davis I, Davis DM. Blood. 2012 Nov 1;120(18):3729-40*

Now added. See also point 4 below for speed of migration on ICAM-1.

Additional references are now cited.

*3) Data showing phosphorylation of specific sites on Vav1 should be provided*.

We have provided data showing site-specific phosphorylation of Y160 of Vav1, a site specifically phosphorylated upon αvβ3 integrin-mediated activation (23) when Zap70 catalytic activity is specifically inhibited (Figure 4).

*4) The main difference in the time-lapse movie shown in*
Figure 6
*is that whereas the* Zap70^+/-^
*T cell is static, the one that has lost Zap70 activity is moving around. Does Zap70 kinase activity regulate the formation of synapses versus kinases? Is there any significant change in motility between both cell types*?

The reviewers asked whether there is a difference in speed of migration of CTL when Zap70 is catalytically inactive. We have addressed this point directly in Videos 1 and 2 and find the speed to be no different whether Zap70 is active or inactive. Our results therefore differ from previous studies using piceatannol to inhibit Zap70 (22). At the concentration used in those studies, piceatannol can potentially inhibit catalytic activity, by 50% or more, of at least 15 out of 110 kinases tested (http://www.kinase-screen.mrc.ac.uk/screening-compounds/345911). In contrast, the system we present is genetically selective and shows no inhibitory effects on wild type T cells at the inhibitor concentrations used. It is possible that their results arose as a result of inhibition of other target kinases.

Since our original submission the Zap70(AS) mice have been crossed onto the OT-I TCR transgenic background. This allowed us to examine synapse formation using CTL where all TCR ensuring equal strength of recognition between all CTL and targets. This allowed us to follow centrosome polarization in 50 movies from CTL with and without Zap70 catalytic activity (Videos 3 and 4). This revealed that the centrosome does begin to polarize in Zap70 catalytically inactive CTL approaching to within 3-5 μm of the synapse, but then aborts. This is consistent with the fixed imaging and is interesting because it suggests that the integrin mediated signaling is sufficient to trigger the first steps of centrosome polarization.

*5) In their Abstract, the authors write, “The membranes at the synapse are unable to flatten to provide extended contact, and TCR microclusters are unable to coalesce.” Yet, none of their data demonstrate this statement: Neither measurement of TCR/CD3 enrichment at the synapse nor analysis of TCR microcluster movements is performed in this study. This statement should thus be removed from the Abstract and if stated in the paper presented as a hypothesis*.

We have modified this as requested.